# Operator Performance of the Digital Setup Fabrication for Orthodontic–Orthognathic Treatment: An Explorative Study

**DOI:** 10.3390/jcm11010145

**Published:** 2021-12-28

**Authors:** Olivier de Waard, Robin Bruggink, Frank Baan, Hendrikus A. J. Reukers, Ewald M. Bronkhorst, Anne Marie Kuijpers-Jagtman, Edwin M. Ongkosuwito

**Affiliations:** 1Department of Dentistry—Orthodontics and Craniofacial Biology, Radboud University Medical Center, 6500 HB Nijmegen, The Netherlands; robin.bruggink@radboudumc.nl (R.B.); frank.baan@radboudumc.nl (F.B.); edwin.ongkosuwito@radboudumc.nl (E.M.O.); 2Radboudumc 3D Lab, Radboud Institute for Health Sciences, Radboud University Medical Center, 6500 HB Nijmegen, The Netherlands; 3Kindt & Reukers Orthodontists, 6524 CV Nijmegen, The Netherlands; reukers@me.com; 4Department of Dentistry, Radboud Institute for Health Sciences, Radboud University Medical Center, 6500 HB Nijmegen, The Netherlands; ewald.bronkhorst@radboudumc.nl; 5Department of Orthodontics, University Medical Center Groningen, University of Groningen, 9713 GZ Groningen, The Netherlands; a.m.kuijpers-jagtman@umcg.nl; 6Department of Orthodontics and Dentofacial Orthopedics, School of Dental Medicine, Medical Faculty, University of Bern, CH-3010 Bern, Switzerland; 7Faculty of Dentistry, Universitas Indonesia, Jakarta 10430, Indonesia

**Keywords:** orthodontics, CBCT, orthognathic surgery, setup

## Abstract

The purpose of this study was to explore the operator performance of the fabrication of digital orthodontic setups integrated into cone beam computed tomography (CBCT) scans. Fifteen patients who underwent a combined orthodontic–orthognathic surgical treatment were included. The pre-treatment digital dental models and CBCT scans were fused, and four operators made virtual setups twice for all patients. Differences between the virtual setups were calculated by recording tooth crown movement from the pre-treatment model to the virtual setup. To examine performance, Pearson’s correlation coefficients, duplicate measurement errors, and inter-operator differences were calculated. For intra-operator performance, correlation values varied among tooth types, with mean correlation values from 0.66 to 0.83 for the maxilla and 0.70 to 0.83 for the mandible. For inter-operator performance, mean correlation values varied from 0.40 to 0.87 for the maxilla and from 0.44 to 0.80 for the mandible. Rotational mean differences exceeded the range of clinical acceptance (>2 degrees) at 18% for the maxilla and 20.8% for the mandible, and translational mean differences exceeded the range of clinical acceptance (0.6 mm) at 9.7% and 26% for the maxilla and mandible, respectively. The intra- and inter-operator performance of digital orthodontic setup construction for virtual three-dimensional orthognathic planning shows significant errors.

## 1. Introduction

Patients with severe skeletal deformities cannot be treated with orthodontics alone to create a harmonious face and stable occlusion. These patients instead undergo a treatment comprising an orthodontic preparation phase followed by orthognathic surgery [1]. Conventional orthognathic treatment planning comprises model plaster surgery and the use of cephalograms to display the relation among dentition, the facial profile, and the facial skeleton [2]. Two-dimensional simulation of the facial appearance of the surgical outcome, however, will be insufficient because it does not address volumetric changes across all facial structures. With the use of cone beam computed tomography (CBCT) and computer-aided design/computer-aided manufacturing technology, it is possible to add a third dimension to surgical treatment planning, allowing for greater accuracy [2]. The aim of the orthodontic preparation is to deliver optimal decompensated dental arches with correct incisor inclination in relation to the jaws to finish in a stable occlusion in a harmonious profile after the surgery [3]. In contrast to the surgical treatment phase, orthodontic preparation is currently not simulated digitally before the start of treatment.

Digital dental models are essential components of orthodontic diagnosis and treatment planning, allowing evaluation of treatment need and outcome [4]). These models are as accurate as traditional plaster models for orthodontic diagnostics and offer advantages that include ease of storage and retrieval, less risk of damage, and better interoffice transferability [5,6,7,8,9]. In 1956, Kesling [10] suggested that cutting and repositioning the teeth in duplicate plaster dental models of the malocclusion, known as the orthodontic setup, would allow simulation of the results before starting orthodontic treatment. With the digitalization of dental models, it is possible to construct digital orthodontic setups that are as accurate as manual setups [11]. These digital versions also offer the advantage of easy fabrication and presentation of therapeutic alternatives to patients and professionals compared to manual setups [8]. Orthodontic setups can be distinguished as either diagnostic or therapeutic. The diagnostic version is a tool to assist the orthodontist in treatment planning for complex cases, such as asymmetric extractions, congenital missing teeth, or combined orthodontic-surgical treatments. Whereas the therapeutic setup can also serve that purpose, and can be used further for the fabrication of orthodontic appliances, such as clear aligners and customized fixed appliance systems [6].

With the introduction of a new method for integrating digital setups into CBCT scans, orthodontic setups can be integrated into the virtual orthognathic planning. Instead of being fabricated just before the surgery, all of the planning can now be mapped out ahead of orthodontic treatment initiation [12]. A study by Falter et al. [13] revealed that one in seven orthognathic patients (13.5%) underwent a different surgical procedure from that originally planned mainly because of facial aesthetic considerations or a negative overjet that did not change as expected in Class III patients. Other reasons for changing a treatment plan included not gaining sufficient width in the maxilla or having an open bite closure after surgically assisted rapid maxillary expansion (SARME) [13]. Pre-treatment virtual planning of the presurgical orthodontic treatment could lead to fewer treatment plan changes and a more predictable treatment outcome. Planning ahead of the combined orthodontic-surgical treatment also enables comparison between the planned result and final outcome.

Data are limited on the reproducibility of digital setups. In a study comparing the digital setups of six clinicians, measured based on the American Board of Orthodontics Objective Grading System (ABO-OGS) scores, it was concluded that the inter- and intra-operator reliabilities were sufficient for general use of the digital orthodontic setup, although ABO-OGS scores differed significantly among the set-ups [14]. This study lacked some crucial information, as the models were evaluated and graded only using the ABO-OGS scores, which omit absolute differences in individual three-dimensional (3D) tooth displacement. Furthermore, to predict and evaluate the accuracy of the dental and skeletal planning of the orthognathic treatment in 3D, the digital orthodontic setups should be produced in the correct anatomical position in the patient’s craniofacial skeleton. No data were published in literature concerning this feature. The goal of the present study was to evaluate the intra- and inter-operator performance of the fabrication of digital orthodontic setups integrated into the craniofacial skeleton of the patient.

## 2. Materials and Methods

### 2.1. Patients

For this retrospective study, patient data were obtained from the patient archive of the section of the Orthodontics and Craniofacial Biology, Department of Dentistry of the Radboud University Medical Centre, Nijmegen, The Netherlands. Only patients requiring a combined orthodontic–orthognathic surgical treatment were included. Exclusion criteria were presence of a developmental deformity and missing more than one tooth per quadrant. The study was conducted between January and June 2020. Because of the explorative nature of the study design, a sample size calculation is not applicable. All patient data were de-identified prior to analysis, and all participants gave informed consent. The Institutional Review Board of the Radboud University Medical Centre issued an approval for this investigation (2016-2690).

### 2.2. Data Acquisition

For diagnostic purposes, CBCT scans were taken prior to the treatment (field of view, 17 × 23 cm; voxel size, 0.3 mm; scanning time, 17.8 s; 891.4 mGy/cm^2^; KaVo 3d eXam, KaVo Dental GmbH, Biberach/Riss Germany). Data from the CBCT exams were exported into a Digital Imaging and Communications in Medicine (DICOM) format. Directly after the acquisition of the CBCT scans plaster models of the dental arches were acquired. The plaster models were digitized with a laser scanner (R500 3D Dental Laser Scanner, 3Shape^®^, Copenhagen, Denmark).

All digital dental models were exported to Standard Tessellation Language (STL) files. Both the DICOM files from the CBCT scans and the STL files from the dental models were imported into the OrthoAnalyzer™ software (2020-1, 3Shape^®^, Copenhagen, Denmark). The dental models were fused with the CBCT data by a semi-automatic surface-based matching procedure executed by the software (Figure 1).

### 2.3. Orthodontic Setup

Four operators participated in this study: two orthodontic residents, each with 3 years of clinical experience (operators 1 and 2), and two experienced orthodontists (operators 3 and 4). All operators had access to the written treatment plan and patient records including the CBCT scan and the digital dental model for each patient. The operators were not involved in the treatment and were blinded to the treatment outcome. The OrthoAnalyzer software was used to fabricate a virtual orthodontic setup on the pre-treatment dental model. The setup preparation for all dental models was done by one experienced operator (operator 4). This phase consisted of a semi-automatic determination of the upper and lower arch forms, tooth axes, points of rotation, and tooth crown segmentations. All individual tooth crowns were semi-automatically segmented after indication of the mesial and distal contact points and gingival borders. After the preparation phase, each model was copied eight times for a blinded delivery to the four operators. All operators were instructed in how to make the setup and practiced with test data before starting with the study data.

The original treatment plan, and key principles of occlusion acted as guidelines for the operators: correct molar relationship in final jaw position, correct crown angulation, correct decompensated crown inclination, no rotations, no interdental spaces when not part of the treatment plan, an appropriate occlusal plane according to the patient craniofacial skeleton on the CBCT, correct interproximal contact point, a normal overjet and overbite (<4 mm), and aligned midlines with respect to the skeletal midline. Mandibular inter-canine distances were respected and acted as a guide for obtaining the final arch width and shape.

All four operators repeated each setup twice with at least a one-month time interval, resulting in 30 setups for each operator and 120 setups in total.

### 2.4. Calculation of Tooth Movement

All setups and individual tooth crowns were exported as STL files and imported into the in-house-created software, 3DMedX^®^ (3D Lab Radboudumc, Nijmegen, The Netherlands). With 3DMedX^®^, all tooth crowns were translated and rotated from the pre-treatment dental model to the orthodontic setup. At the same time, this movement was recorded automatically and saved for each individual tooth. Six parameters of movements were computed: three rotational (yaw, roll, and pitch; Figure 2) and three translational (left to right (X), anterior to posterior (Y), and cranial to caudal (Z). Together, these parameters form the six degrees of freedom.

### 2.5. Statistical Analysis

All data were imported into IBM SPSS software, version 26.0 (IBM, Armonk, NY, USA). Paired *t*-tests were used to compare the differences between the pre-treatment models and orthodontics setups within and between operators. Intra- and inter-operator performance of the method was expressed by Pearson’s correlation coefficients and duplicate measurement errors (DME). The differences expressing the systematic errors between operators were expressed by mean differences and confidence intervals and tested with the paired sample *t*-test. The inter-operator performance was calculated on the first setup series. For an assessment of the relevance of the inter-operator differences, tolerance levels were set at 0.6 mm for translations and 2 degrees for rotations. The significance level was set at 5%. A correlation of 0.90 or higher is considered excellent, between 0.75 and 0.90 is good, between 0.50 and 0.75 is moderate, and less than 0.50 is considered poor [15].

## 3. Results

### 3.1. Patients

Fifteen skeletally mature non-syndromic patients who underwent orthodontic treatment combined with orthognathic surgery were included. Treatment was initiated from 2013 to 2015. Ten patients had a Class II malocclusion with mandibular hypoplasia, and five patients had a Class III malocclusion with mandibular hyperplasia and/or maxillary hypoplasia.

Fourteen patients were prepared for bimaxillary (bimax) surgery and one for a LeFort 1 osteotomy. Seven of the fourteen patients were prepared for a SARME procedure as well as a bimax surgery. Seven patients had extraction therapy to align the teeth, consisting of two premolar extractions in the lower arch for all of them and extraction of two premolars in the upper arch for three.

### 3.2. Intra-Operator Performance

To assess the intra-operator performance of fabrication of the orthodontic setups, the first and second setups of each operator for every patient were compared with each other based on a match with the underlying CBCT (Figure 3). Figure 4 shows the correlation coefficients for all translations and rotations and the correlation coefficients of all tooth movement displacements for all four tooth types separately, for the maxilla and mandible, respectively. These values represent the differences between the tooth movement displacements between the first and second orthodontic setups for all operators together. No specification is shown in the figures for patients and operators individually because no relevant differences concerning the intra-operator performance was detected between patients or between operators.

In the maxilla, the incisors (Figure 4A) showed the highest correlations, with a mean of 0.83 (range 0.52–0.95), and the molars the lowest, with a mean of 0.66 (range 0.28–0.93). Among the translations and rotations, the left–right displacements, as expressed by X, showed the highest correlation, with a mean of 0.87 (range 0.75–0.96), and roll showed the lowest, with a mean of 0.63 (range 0.14–0.94).

Good correlation was found for the mandibular canines (Figure 4B) (mean 0.83, range 0.59–0.95). In the mandible, the molars showed the lowest correlation values (mean 0.70, range 0.49–0.95). Among the six parameters of movements, X values had the lowest correlations (mean 0.67, range 0.44–0.83), and Y showed the highest correlations (mean 0.85, range 0.51–0.95).

Figure 4C shows the intra-operator DME for orthodontic setup construction. Reproducibility depends on the type of tooth displacement, and the DME for the vertical displacement as expressed by Z was the smallest in the maxilla (0.40, range 0.25–0.62) and mandible (mean 0.43, range 0.32–0.62). The anterior–posterior movements, as expressed by Y, had the highest DME both in the maxilla (mean 0.56, range 0.29–0.84) and mandible (mean 0.70, range 0.45–1.37). For rotational displacements, roll showed the lowest DME in the maxilla (mean 1.97, range 0.59–2.78) and mandible (mean 2.31, range 1.24–3.44) and yaw the highest in the maxilla (mean 2.99, range 1.54–3.99) and mandible (mean 3.67, range 2.27–4.87). No clinically relevant differences were seen among individual tooth types concerning the DME for the intra-operator performance (not shown).

### 3.3. Inter-Operator Performance

#### 3.3.1. Correlation and DME

Figure 5 shows the inter-operator performance (Pearson correlation coefficients) among all four operators for all translations, rotations, and all tooth types separately for the maxillary (Figure 5A) and mandibular (Figure 5B) arches. For the maxilla (Figure 5A), the left–right displacements, as expressed by X, showed the best correlation (mean 0.87, range 0.75–0.95). The vertical displacements, as expressed by Z, had the lowest correlations (mean 0.40, range −0.01–0.78). For the mandible, yaw had the best correlation values (mean 0.80, range 0.63–0.93) and Z the lowest (mean 0.44, range −0.01–0.80).

In the maxilla, the operators had the best agreement for displacements of incisors and the worst for molars (Figure 5A). In the mandible, they had the best agreement for canines and the worst for the molars.

Figure 5C shows the inter-operator DME of tooth displacements during orthodontic setup construction. For translational and rotational displacements, the parameters X and roll respectively had the lowest DME. In contrast, the DME for Y in the mandible was higher (mean 0.87 mm, range 0.65–1.27 mm), and the yaw in the lower jaw was relatively high (mean 4.33, range 2.79–6.31). No clinically relevant differences were seen among individual tooth types concerning the DME for the inter-operator performance (not shown).

#### 3.3.2. Occurrence of Large Inter-Operator Differences in Translations

Mean differences for translations larger than 0.6 mm were found for 7 of the 72 (12 variables, 6 pairwise comparisons between operators) inter-operator differences in the maxilla and 18 of the 72 inter-operator differences in the mandible. Table 1 shows these out-of-range mean differences with corresponding DME values, confidence intervals, and *p* values. All inter-operator differences greater than 0.6 mm for translational movements were statistically significant.

#### 3.3.3. Occurrence of Large Inter-Operator Differences in Rotations

Mean differences in the maxilla for rotations larger than 2 degrees were found in 13 of 72 differences. The largest was the yaw of incisors between operators 2 and 3, with a value of −4.87 degrees (Table 2).

Mean differences for translations larger than 2 degrees in the mandible were found in 15 of 72 differences. Pitch showed the worst inter-operator agreement, with a difference > 2 degrees in the mandible 10 out of 24 times (4 tooth type variables, 6 pairwise comparisons between operators) (Table 2). The largest mean difference was found for the pitch of incisors between operators 1 and 4, with a value of 5.72 degrees.

With exception of the comparison between operators 2 and 3 for yaw in the mandible for premolars and canines all values were statistically significant.

#### 3.3.4. Influence of Extraction and SARME Therapy

Table 3 and Table 4 show data representing the influence of premolar extraction and SARME therapy on the reproducibility of the orthodontic setups. Significant differences were found for the random error between extraction and non-extraction cases for parameters X and Y, expressing the left–right and anterior–posterior displacements, respectively (Table 3). Significant differences were also found between the SARME and non-SARME groups for the parameters roll, yaw, and Z for molars in the mandible and for the yaw of incisors in the upper arch (Table 4). An overview of all the data concerning the influence of extraction and SARME therapy, non-significant data included, is provided as a Appendix A.

## 4. Discussion

Limited information is available in the literature concerning the performance of diagnostic orthodontic setup fabrication for orthodontic–orthognathic surgery planning. Published studies are equally scarce regarding application of the therapeutic setup in customized bracket or clear aligner systems such as Incognito™, Insignia™, or Invisalign^®^. Fabels et al. evaluated the intra- and inter-operator performance of six clinicians using the ABO-OGS score to evaluate set-up fabrication using the OrthoCAD software [14]. In comparing setups, the intra-examiner mean absolute differences in total ABO-OGS scores varied statistically significantly between 2.17 and 6.00 points. Interexaminer mean absolute differences varied statistically significantly between 4.77 and 5.56 points. Reliability showed significant good (ICC, 0.6–0.8) agreement. A study by Barreto et al. [8] compared the reproducibility of digital setups with manual setups of 20 patients and concluded that both can be reliably reproduced. In this study, no significant differences were found between intermolar, intercanine, and mandibular arch length measurements between digital and manual dental set-ups and the final dental model after orthodontic treatment. The correlations for most measurements were outstanding (r = 0.75 to 0.93). However, in both abovementioned studies, orthognathic cases were excluded, and setups were not made in relation to the facial skeleton using CBCT, which is essential in the planning of an orthognathic case. In addition, no 3D measurements were performed to describe the deviations.

The results for the intra-operator performance in the present study revealed a moderate correlation (>0.5) for the maxillary and mandibular molars (Figure 4A,B). Especially, the roll in the maxilla and the X tooth displacements in the mandible showed lower correlation values. This finding could be relevant because different roll and X values for molars can lead to differences in the vertical and transversal dimension of the bite and to a different presurgical outcome. In the mandible, the reproducibility of the tooth positions was low for the yaw, as indicated by the duplicate measurement errors. These differences were mainly the result of difficulties for the operators in reproducing the torque and angulation, as expressed by the pitch and roll and translational displacements (X) in the mandible in case of the molars, and the mesiodistal rotation in general. These differences might have resulted from the absence of strict norms for these parameters when producing a setup. However, it is arguable whether deviations of a few degrees in angulation or torque of the molars would affect the surgical outcome. The results for the inter-operator performance revealed differences between operators for the translations on the vertical plane and rotations on the horizontal axis, with low correlations for the Z values and pitch (Figure 5A,B), especially for maxillary molars. As also seen for the results of the intra-operator performance, the yaw showed the highest values for the DME, indicating a relatively large random error.

For an assessment of the clinical relevance of inter-operator differences, tolerance levels must be considered. These are subjectively described in the literature. Larson et al. [16] used 0.5 mm for translational differences and 2 degrees for rotational differences as clinically relevant difference levels. Grauer and Proffit considered average translational discrepancies of <1 mm or <4 degrees for rotational discrepancies as clinically irrelevant in their studies comparing the setup with the clinical outcome after lingual orthodontic therapy [17]. Based on these studies, the arbitrary limits of 0.6 mm for translations and 2 degrees for rotations are chosen as tolerance levels. The results in this study were based on exceptions to these ranges of clinical acceptability. Differences exceeding the ranges were found in 9.7% (7/72) of the translational mean differences and in 18% (13/72) of the rotational mean differences in the maxilla (Table 1 and Table 2). For the mandible, 26.3% (18/72) of the translational mean differences and 20.8% (15/72) of the rotational mean differences exceeded the range of clinical acceptance (Table 1 and Table 2). Notable for exceeding this cutoff were sagittal and vertical positioning of lower incisors and canines, expressed by Y and Z (Table 1). For rotations, incisors and canines showed relatively frequent large mean differences for yaw and pitch (Table 2).

A possible explanation for these discrepancies is that the treatment outcome was susceptible to the subjective judgment of the clinician. Sometimes, occlusion and esthetics are in tension, leading to different decisions by different clinicians favoring one over the other [14].

This study included only combined orthodontic–orthognathic treatment cases, which are considered the more difficult cases. This could partly explain the differences in operator performance compared to the results of previous studies [8,14]. Differences in clinical experience between operators may have influenced the results. Two operators were postgraduate students in their final year of graduation, and the other two operators were experienced orthodontists with possibly different insights into tooth displacement feasibility. It might be possible as well that the level of experience of the operators with the sophisticated OrthoAnalyzer™ software played a role. To reduce the influence of unequal experience with the software, only the most experienced operator performed the setup preparation for all dental models (see Section 2.3).

The differences between and within operator can be partly explained by the influence of extractions. For all extraction patients, extractions of the lower premolars were planned. The influence of tooth extractions on the reproducibility is mainly seen for the Y and X parameters in the lower jaw (Table 3). It seems rather difficult to virtually estimate the closing of the extraction diastema’s in a reproducible way in the sagittal and transversal plane. In addition, SARME may have played a role. Significant errors for SARME therapy were seen for molars in the lower jaw for vertical, torque, and rotational displacements. Rotational movements of incisors in the upper arch also showed significant differences in random errors (Table 4). All statistically significant errors were associated with larger standard errors for SARME or extraction cases compared to non-extraction or non-SARME cases, respectively. It is complicated to keep track of tooth movement changes during the setup fabrication in the software, which easily introduces disorientation in complex cases, such as with extractions and SARME. The result can be more variation and less reproducibility of tooth displacement estimations. A lack of strict rules about closure of extraction diastemas in a setup also may lead to variation during setup preparation.

Another limitation of this study is the choice for the explorative design because of the limited amount of available literature and the use of a relatively new 3D method to follow tooth movements in 3D [12]. This may complicate the judgment of the results in a clinical perspective. However, the results of this explorative study can be adequately used to optimize the set-up fabrication process and to design follow-up studies.

The inter-operator variability in this study indicates the need for a protocol for the simulation of planned tooth movements in orthodontic setups during treatment planning for combined orthodontic-surgical treatment. Software improvements indicating evidence-based biological limitations of tooth displacement during the virtual setup production could help improve the performance of orthognathic planning at the start of treatment. More research comparing virtual setups with the actual outcome by matching on the skull base could provide information about preventing such planning errors and will leave less chance for the subjective interpretation of the orthodontist, allowing for more predictable planning.

## 5. Conclusions

The intra- and inter-operator performance of digital orthodontic setup construction for virtual orthognathic planning in 3D show significant errors. Further research is needed to objectify the procedure, while software enhancements could improve orientation and limits of tooth displacements during the process.

## Figures and Tables

**Figure 1 jcm-11-00145-f001:**
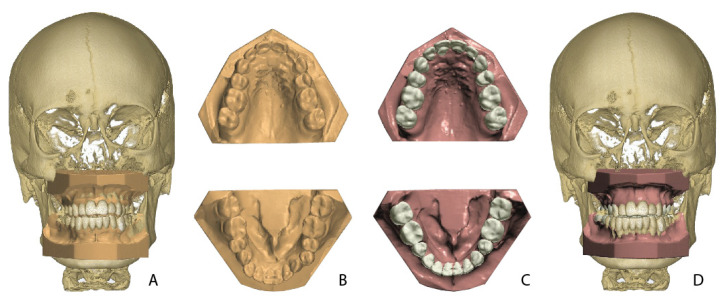
Overview of models in the setup fabrication process. (**A**) Pre-treatment dental model in CBCT. (**B**) Pre-treatment model. (**C**) Orthodontic setup. (**D**) Orthodontic model in CBCT.

**Figure 2 jcm-11-00145-f002:**
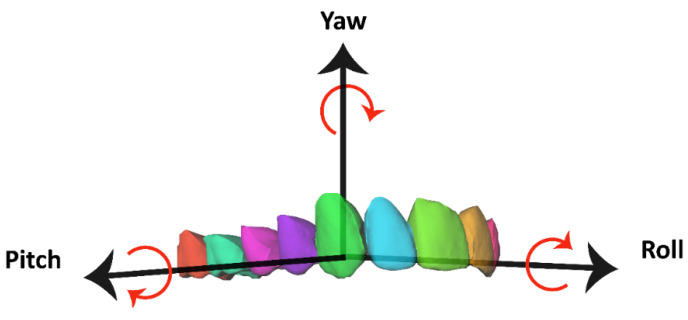
Teeth can move around three rotation axes—yaw, roll, and pitch—as shown in an upper dental arch.

**Figure 3 jcm-11-00145-f003:**
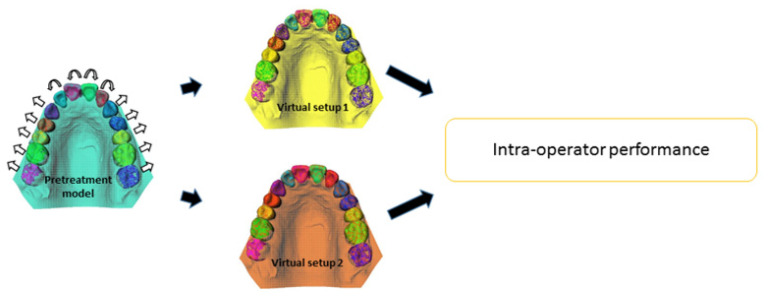
During the fabrication of the orthodontic setup, all individual segmented tooth crowns, displayed in different colors, can be moved in any direction, as shown by the white arrows. This procedure is repeated by the same operator a second time. The tooth movements of the first and second setups are subsequently compared to assess the intra-operator performance.

**Figure 4 jcm-11-00145-f004:**
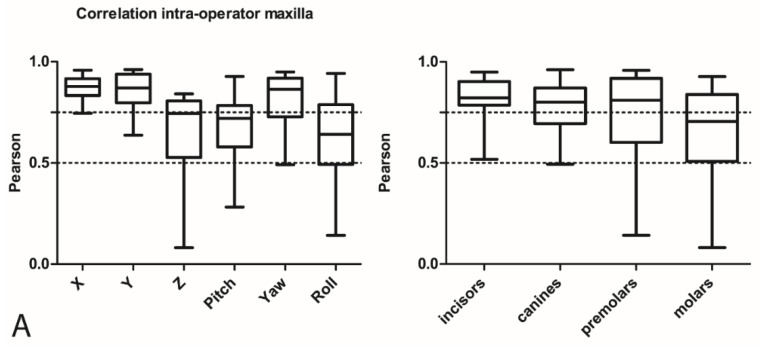
(**A**) Intra-operator correlations in the maxilla for translations, rotations, and tooth types. (**B**) Intra-operator correlations in the mandible for translations, rotations, and tooth type. (**C**) Intra-operator DME for all translational and rotational movements.

**Figure 5 jcm-11-00145-f005:**
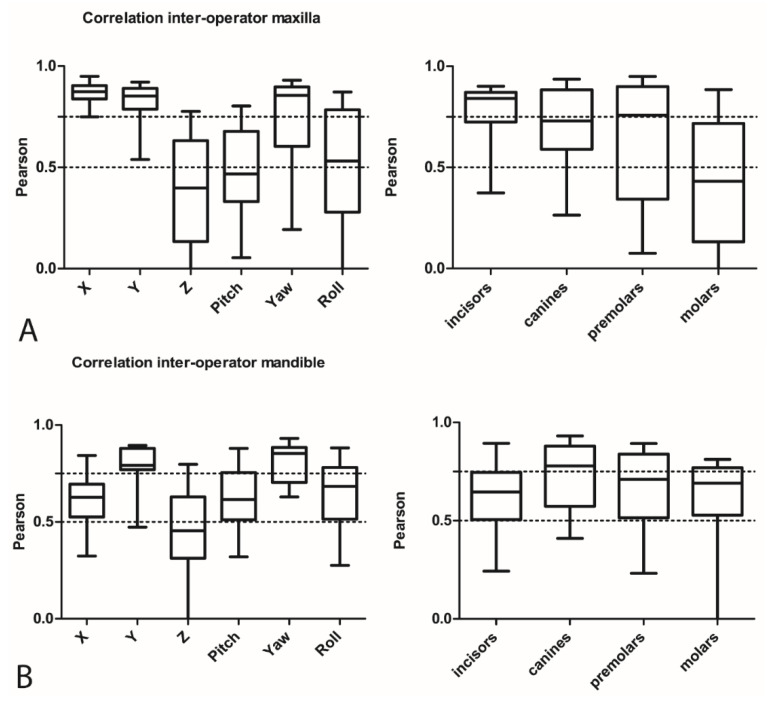
(**A**) Inter-operator correlations in the maxilla for rotations, translations, and tooth types. (**B**) Inter-operator correlations in the mandible for rotations, translations, and tooth types. (**C**) Inter-operator DME for all translational and rotational movements.

**Table 1 jcm-11-00145-t001:** Inter-operator mean differences, 95% confidence intervals (CIs), DME values, and result of the paired sample *t* test (*p*-value) for differences in translations in maxilla and mandible larger than 0.6 mm.

Jaw	Tooth Type	Parameter	Inter-Operator	N	DME	Mean Difference (mm)	95% CI		*p*
							Lower	Upper	
Mx	I	Z	2–3	60	0.714	−0.655	−0.92	−0.40	<0.001
Mx	I	Z	2–4	60	0.604	−0.836	−1.06	−0.62	<0.001
Mx	PM	X	1–2	49	0.493	0.837	0.64	1.04	<0.001
Mx	PM	X	2–3	49	0.455	0.828	0.64	1.01	<0.001
Mx	PM	X	2–4	49	0.587	0.868	0.63	1.11	<0.001
Mx	M	X	1–2	59	0.694	0.815	0.56	1.07	<0.001
Mx	I	Y	1–4	60	0.672	0.616	0.37	0.86	<0.001
Md	I	Y	1–2	60	0.837	−1.119	−1.42	−0.81	<0.001
Md	I	Y	2–4	60	1.131	−1.117	−1.53	−0.70	<0.001
Md	I	Y	1–3	60	0.723	0.628	0.36	0.89	<0.001
Md	I	Y	3–4	60	0.832	−0.625	−0.93	−0.32	<0.001
Md	C	Y	1–2	30	0.84	−0.614	−1.06	−0.17	0.008
Md	C	Y	2–4	30	1.265	−0.976	−1.64	−0.31	0.006
Md	M	Y	2–3	55	0.906	−0.724	−1.07	−0.38	<0.001
Md	M	Y	2–4	55	0.994	−0.709	−1.09	−0.33	<0.001
Md	I	Z	1–2	60	0.572	0.764	0.56	0.97	<0.001
Md	I	Z	1–4	60	0.781	−1.239	−1.53	−0.95	<0.001
Md	I	Z	3–4	60	0.725	−0.718	−0.98	−0.45	<0.001
Md	C	Z	1–2	30	0.528	0.642	0.36	0.92	<0.001
Md	C	Z	1–4	30	0.604	−0.71	−1.03	0.39	<0.001
Md	PM	Z	1–2	44	0.482	0.621	0.41	0.83	<0.001
Md	PM	Z	2–4	44	0.727	0.657	0.34	0.97	<0.001
Md	PM	Z	2–3	44	0.559	0.610	0.37	0.85	<0.001
Md	M	Z	2–3	55	0.721	1.152	0.88	1.43	<0.001
Md	M	Z	2–4	55	0.982	1.451	1.08	1.83	<0.001

N = available teeth × number of set-ups. Mx maxilla, Md mandible. I incisors, C canines, PM premolars, M molars. X left-right translation, Y anterior-posterior translation and Z cranial-caudal translation. DME duplicate measurement error. CI confidence interval. Significance was set at α = 0.05.

**Table 2 jcm-11-00145-t002:** Inter-operator mean differences, 95% confidence interval (CI), DME values, and result of the paired sample *t* test (*p* value) for differences in rotations larger than 2 degrees in the maxilla and mandible.

Jaw	Tooth Type	Parameter	Inter-Operator	N	DME	Mean Difference (Degrees)	95% CI		*p*
							Lower	Upper	
Mx	M	Pitch	2–4	59	3.28	2.10	0.90	3.31	0.001
Mx	M	Pitch	1–3	59	5.10	3.62	1.74	5.50	<0.001
Mx	M	Pitch	1–4	59	5.00	3.85	2.00	5.69	<0.001
Mx	PM	Roll	1–2	49	2.90	3.57	2.39	4.75	<0.001
Mx	PM	Roll	2–3	49	2.64	3.04	1.97	4.11	<0.001
Mx	PM	Roll	2–4	49	2.91	3.67	2.49	4.85	<0.001
Mx	I	Yaw	1–2	60	3.36	2.07	0.84	3.30	0.001
Mx	I	Yaw	2–3	60	3.81	−2.80	−4.19	−1.41	<0.001
Mx	I	Yaw	1–3	60	3.74	−4.87	−6.24	−3.51	<0.001
Mx	I	Yaw	3–4	60	3.08	3.22	2.10	4.34	<0.001
Mx	C	Yaw	2–3	30	4.89	−3.10	−5.68	−0.52	0.020
Mx	C	Yaw	2–4	30	5.03	−4.56	−7.22	−1.90	0.001
Mx	C	Yaw	1–4	30	4.34	−2.75	−5.04	−0.46	0.020
Md	I	Pitch	2–3	60	3.14	3.14	1.99	4.28	<0.001
Md	I	Pitch	2–3	60	4.02	4.45	2.98	5.92	<0.001
Md	I	Pitch	1–3	60	3.06	4.41	3.29	5.53	<0.001
Md	I	Pitch	1–4	60	3.73	5.72	4.36	7.09	<0.001
Md	C	Pitch	2–3	30	4.60	4.35	1.92	6.78	0.001
Md	C	Pitch	2–3	30	4.70	5.64	3.16	8.12	<0.001
Md	C	Pitch	1–3	30	3.84	3.85	1.82	5.88	0.001
Md	C	Pitch	1–4	30	4.05	5.15	3.01	7.29	<0.001
Md	PM	Pitch	2–4	44	2.74	2.78	1.60	3.96	<0.001
Md	PM	Pitch	1–4	44	2.53	2.43	1.34	3.51	<0.001
Md	C	Yaw	2–3	30	5.70	−2.30	−5.31	0.71	0.129
Md	C	Yaw	1–3	30	5.53	−2.78	−5.70	0.14	0.061
Md	PM	Yaw	2–3	44	6.31	−2.31	−5.03	0.40	0.092
Md	M	Yaw	2–4	55	2.95	2.03	0.90	3.16	0.001
Md	M	Yaw	3–4	55	2.83	2.46	1.38	3.54	<0.001

N = available teeth × number of set-ups. Mx maxilla, Md mandible. I incisors, C canines, PM premolars, M molars. X left-right translation, Y anterior-posterior translation and Z cranial-caudal translation. DME duplicate measurement error. CI confidence interval. Significance was set at α = 0.05.

**Table 3 jcm-11-00145-t003:** Influence of extraction therapy on the fabrication of the setup. Only significant differences with corresponding parameter are shown.

Jaw	Tooth Type	Parameter	Mean of Differences in Standard Error (mm)	95% CI		*p* Value
				Lower	Upper	
Maxilla	Canines	Y	0.35	0.07	0.63	0.028
Maxilla	Premolars	Y	0.38	0.22	0.54	0.004
Mandible	Incisors	Y	0.42	0.23	0.61	0.006
Mandible	Canines	Y	0.66	0.39	0.93	0.004
Mandible	Premolars	X	0.47	0.19	0.75	0.013
Mandible	Premolars	Y	0.84	0.03	1.65	0.046
Mandible	Molars	X	0.50	0.16	0.83	0.018

X left-right translation, Y anterior-posterior translation, CI confidence interval, Significance was set at α = 0.05.

**Table 4 jcm-11-00145-t004:** Influence of SARME therapy on the fabrication of the setup. Only significant differences with corresponding parameter are shown.

Jaw	Tooth Type	Parameter	Mean of Differences in Standard Error	95% CI		*p* Value
				Lower	Upper	
Mandible	Molars	Roll	1.97°	0.37	3.57	0.030
Mandible	Molars	Yaw	0.74°	0.23	1.25	0.019
Mandible	Molars	Z	0.25 mm	0.01	0.40	0.045
Maxilla	Incisors	Yaw	1.05°	0.17	1.93	0.032

Z cranial-caudal translation, CI confidence interval, significance was set at α = 0.05.

## Data Availability

The data presented in this study are available on request from the corresponding author. The data are not publicly available due to privacy reasons.

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
