# Peer review of "Operator Performance of the Digital Setup Fabrication for Orthodontic–Orthognathic Treatment: An Explorative Study"

_jcm, 2021, doi:10.3390/jcm11010145_

Round 1
Reviewer 1 Report
I have read and reviewed with great interest the manuscript entitled "Operator performance of the digital setup fabrication for orthodontic-orthognathic treatment: an explorative study".
The authors should review and respond to the following points:
1) In Material and Method the authors do not indicate on what date they conducted the study.
2) The authors should describe a conclusion section at the end of the manuscript.
3) The authors should review the discussion section in detail. I do not see that the authors have adequately compared their results with the results of other authors.
4) What are the limitations of the study?
Author Response
Thank you for reviewing the manuscript. The remarks are very valuable and supported us to improve and revise the text.
Comment 1:
In Material and Method the authors do not indicate on what date they conducted the study.
Response 1:
Thank you for your observation. We agree and have corrected this in the revised version. ‘The study was conducted between January and June 2020’.
Comment 2:
The authors should describe a conclusion section at the end of the manuscript.
Response 2:
In the revised version we have added a conclusion at the end of the discussion section: “The intra- and inter-operator performance of digital orthodontic setup construction for virtual orthognathic planning in 3D shows significant errors. Further research is needed to objectify the procedure, while software enhancements could improve orientation and limits of tooth displacements during the process.”
Comment 3:
The authors should review the discussion section in detail. I do not see that the authors have adequately compared their results with the results of other authors.
Response 3:
We agree with you that this would be a valuable improvement of the article. Unfortunately to the best of our knowledge very few comparable studies are available in literature. This was also one of reasons we have chosen for the explorative design of the study. We compared our results with the studies of Fabels [1] and Barreto [2], and we have added now more information about both studies in the beginning of the discussion section of the revised version of the manuscript.
Comment 4:
What are the limitations of the study?
Response 4:
We agree that our discussion and conclusions do not completely cover all the limitations, we have revised our article accordingly (end of the discussion section): “Another limitation of this study is the choice for the explorative design because of the limited amount of available literature and the use of a relatively new 3D method to follow tooth movements in 3D [12]. This may complicate the judgment of the results in a clinical perspective. However the results of this explorative study can be adequately used to optimize the set-up fabrication process and to design follow-up studies. ”
Reviewer 2 Report
Thank you very much to give the opportunity to review this work.
The authors should provide the use of personal pronouns such as we, our etc, overall the manuscript. Please see the lines 96, 376 and 377.
Please discuss the possible complications of conventional ortognathic surgical interventions by citing following articles:
Gulses A, Aydintug YS, Sencimen M, Bayar GR, Acikel CH. Evaluation of neurosensory alterations via clinical neurosensory tests following anterior maxillary osteotomy (Bell technique). Int J Oral Maxillofac Surg. 2012 Nov;41(11):1353-60. doi: 10.1016/j.ijom.2012.03.021. Epub 2012 Apr 23. PMID: 22534359.
and
Gulses A, Kilic C, Sencimen M. Determination of a safety zone for transbuccal trocar placement: an anatomical study. Int J Oral Maxillofac Surg. 2012 Aug;41(8):930-3. doi: 10.1016/j.ijom.2012.02.013. Epub 2012 Mar 21. PMID: 22440614.
Best regards
Author Response
Thank you for reviewing the article. The remarks are very valuable and supported us to improve and revise the text.
Comment 1:
The authors should provide the use of personal pronouns such as we, our etc, overall the manuscript. Please see the lines 96, 376 and 377.
Response 1:
Thank you for your observation, we have corrected this.
Comment 2:
Please discuss the possible complications of conventional ortognathic surgical interventions by citing following articles:
Please discuss the possible complications of conventional ortognathic surgical interventions by citing following articles:
Gulses A, Aydintug YS, Sencimen M, Bayar GR, Acikel CH. Evaluation of neurosensory alterations via clinical neurosensory tests following anterior maxillary osteotomy (Bell technique). Int J Oral Maxillofac Surg. 2012 Nov;41(11):1353-60. doi: 10.1016/j.ijom.2012.03.021. Epub 2012 Apr 23. PMID: 22534359.
and
Gulses A, Kilic C, Sencimen M. Determination of a safety zone for transbuccal trocar placement: an anatomical study. Int J Oral Maxillofac Surg. 2012 Aug;41(8):930-3. doi: 10.1016/j.ijom.2012.02.013. Epub 2012 Mar 21. PMID: 22440614.
Response 2:
Thank you for this suggestion. Both articles are very interesting. However possible complications of orthognathic surgery are beyond the scope of our research. Therefore, we decided not to include these two references.
Reviewer 3 Report
Today, orthodontic as well as orthognathic surgery treatment planning is more and more managed using a “digital workflow”. 3D CBCTs of the skeleton and digital models of the teeth are virtually fused and treatment planning is started with a virtual set-up of the intended tooth movements which is the prerequisite for the subsequent relocation of the skeletal bases of the maxilla and mandible into the desired new positions. Later on, after the pre-surgical orthodontic phase, actual records are prepared, and with the help of CAM-technique, surgical splints are fabricated which are used to transfer the treatment plan into the operation room.
In their study, the authors intended to explore the precision of the first step of this procedure, namely the intra- and inter-operator error of the virtual set-up. For this purpose, calculations of Correlation coefficients and Duplicate measurement errors were performed separately for different groups of teeth.
Major concerns
- The study was designed as a pure descriptive investigation without having a concrete hypothesis to start with. Thus, authors state that :” Because of the explorative nature of the study design a sample size calculation is not applicable”. As a consequence, it becomes rather difficult for the reader to judge the huge number of statistical data with respect to their clinical meaning. In addition, only few “mechanistical” explanations for the variability of the results are offered (“extraction” or SARPE” treatment).
- For this reviewer, the chosen “tolerance levels” are not comprehensible. The point of discussion is the relevance of the quantity of the “planning errors” in relation to the errors related to the true clinical outcome following pre-surgical orthodontic treatment and to the transfer of the treatment plan into the operation room.
- The presented results appers to be very much operator-dependent. Thus, generalizability seems questionable. In this context, the granularity of the treatment plan that was presented to the test persons does not become obvious (f.g. details about the sagittal and vertical positions of the incisors etc.).
Author Response
Comment 1:
The study was designed as a pure descriptive investigation without having a concrete hypothesis to start with. Thus, authors state that :” Because of the explorative nature of the study design a sample size calculation is not applicable”. As a consequence, it becomes rather difficult for the reader to judge the huge number of statistical data with respect to their clinical meaning. In addition, only few “mechanistical” explanations for the variability of the results are offered (“extraction” or SARPE” treatment).
Response 1:
This study was not designed to reach a prespecified level of precision in knowledge on the performance of this digital set-up routine. For such a study to be designed, one needs to have knowledge on error levels and the associated variability that is, or better was, unavailable. By this descriptive approach that knowledge gap is now for a large part fulfilled. So a subsequent study can be designed that can utilize this knowledge. This can serve various goals. By looking at the results of this explorative study it can give valuable information where to improve the set-up procedure and it can give information on various forms of variability so a subsequent study can lean on a solid sample size calculation. It might even give information for a more efficient design of the study, for example by focusing on measurements and analysis of the most vulnerable outcomes. To summarize this: the value of this study is not to supply data that can be judged. It helps to decide whether to adopt or reject this technique. Its results are an auxiliary step, necessary both for optimization of the technique as well as for designing follow-up studies.
To make this limitation clear to the reader we have added additional information concerning this in the discussion and we extended the discussion with more limitations.
Furthermore we have added conclusions in which we emphasize several reasons to clarify the variability that was found. See conclusions in manuscript.
Comment 2: For this reviewer, the chosen “tolerance levels” are not comprehensible. The point of discussion is the relevance of the quantity of the “planning errors” in relation to the errors related to the true clinical outcome following pre-surgical orthodontic treatment and to the transfer of the treatment plan into the operation room
Response 2:
We thank the reviewer for this comment, we agree that in the end it is the consequence of planning errors in set-up routine on the eventual clinical outcomes that really matter. In terms of pharmacological research, that would ask for a phase 3 clinical trial. But we think that before starting such an undertaking, a serious effort has to be made in phase 2 studies (to stick to the pharmacological analogy). The cut-offs used are just tools to help present and organize the large amount of information this study delivered. They also make the comparison with other articles possible. By no means are they to be taken as a boundary between clinically relevant and irrelevant errors.
Comment 3:
The presented results appears to be very much operator-dependent. Thus, generalizability seems questionable. In this context, the granularity of the treatment plan that was presented to the test persons does not become obvious (f.g. details about the sagittal and vertical positions of the incisors etc.).
Response 3:
The outcome is indeed operator-dependent, this is one of our main findings. All our operators had access to the complete treatment plan and patient records but not to the results of the combined orthodontic/surgical procedure of course. By purpose we did not inform the operators in the study on specific details as vertical positions of incisors because this would make the set-up fabrication less comparable to daily life in which orthodontists’ personal preferences will differ.
Round 2
Reviewer 3 Report
The authors answered my questions and added some aspects related to the limitations of the study design to their discussion.
If the journal is fine with this explorative kind of research design, I do not have any fiurther concerns.